# Comparative Efficacy of Subcutaneous Versus Intravenous Interleukin 12/23 Inhibitors for the Remission of Moderate to Severe Crohn’s Disease: A Systematic Review and Meta-Analysis

**DOI:** 10.3390/biomedicines13030702

**Published:** 2025-03-12

**Authors:** Nouran Alwisi, Rana Ismail, Hissa Al-Kuwari, Khalifa H. Al-Ansari, Mohammed A. Al-Matwi, Noor A. Aweer, Wejdan N. Al-Marri, Yousif Al-Kubaisi, Muneera Al-Mohannadi, Shahd Hamran, Suhail A. R. Doi, Habib H. Farooqui, Tawanda Chivese

**Affiliations:** 1College of Medicine, QU Health, Qatar University, Doha P.O. Box 2713, Qatar; na2104730@qu.edu.qa (N.A.); ri2108210@qu.edu.qa (R.I.); ha2104524@qu.edu.qa (H.A.-K.); ka1801508@qu.edu.qa (K.H.A.-A.); ma2104272@qu.edu.qa (M.A.A.-M.); na2003836@qu.edu.qa (N.A.A.); wa1903516@qu.edu.qa (W.N.A.-M.); ya1808493@qu.edu.qa (Y.A.-K.); sh2004556@qu.edu.qa (S.H.); sdoi@qu.edu.qa (S.A.R.D.); 2Department of Gastroenterology and Hepatology, Hamad Medical Corporation, Doha P.O. Box 3050, Qatar; malmohannadi@hamad.qa

**Keywords:** Crohn’s disease (CD), interleukin inhibitors, monoclonal antibodies (mAbs), subcutaneous (SC), intravenous (IV), meta-analysis

## Abstract

**Background/Objectives**: Interleukin 12/23 inhibitors are a newer class of monoclonal antibodies used to induce and maintain remission for Crohn’s disease (CD), a chronic inflammatory bowel disease, when patients do not respond to conventional immunomodulatory drugs or first-line monoclonal antibody therapies. Although biologics are best administered intravenously, subcutaneous administration has been trialed, with mixed results. This research synthesized evidence on the efficacy and safety of subcutaneous compared to intravenous administration of interleukin 12/23 inhibitors for moderate to severe CD. **Methods**: In this systematic review and meta-analysis, we searched Cochrane, PubMed, SCOPUS, CINHAL, and preprint archives for randomized controlled trials (RCTs) that compared the efficacy and safety of subcutaneous to intravenous interleukin 12/23 inhibitors for the remission of CD. After study quality assessment, a meta-analysis was carried out using a bias-adjusted inverse variance heterogeneity model, heterogeneity was assessed using I^2^, and publication bias was performed using Doi plots. Evidence certainty was assessed using Grading of Recommendations, Assessment, Development, and Evaluations (GRADE). **Results**: Seven RCTs, with 2179 participants, all with moderate to severe CD, were included. After meta-analysis, subcutaneous compared to intravenous administration showed similar efficacy for the induction of remission (OR 0.77, 95%CI 0.53–1.12), with no-to-low heterogeneity (I^2^ = 0%, *p* = 0.97). For the maintenance of remission, only two studies had analyzable data, and they showed that subcutaneous interleukin 12/23 inhibitors were equal or better compared to intravenous administration. Further syntheses showed that subcutaneous compared to intravenous administration of interleukin 12/23 inhibitors had almost similar odds of adverse events (OR 0.91, 95%CI 0.63–1.32, I^2^ = 39%), serious adverse events (OR 0.97, 95%CI 0.61–1.53, I^2^ = 0%), and treatment discontinuation (OR 1.06, 95%CI 0.67–1.68, I^2^ = 0%). **Conclusions**: In individuals with moderate to severe CD, subcutaneous administration has similar efficacy for inducing remission with comparable safety. More RCTs are needed to confirm these findings.

## 1. Introduction

Crohn’s disease (CD) is a chronic, disabling, and progressive inflammatory bowel disease (IBD) of the gastrointestinal tract [1]. Once classified as rare, the prevalence of IBD, which CD belongs to, is increasing in step with the rapid epidemiological transitions occurring in low- and middle-income countries (LMIC) [2,3,4]. Between 1990 and 2019, the number of people affected by IBD increased from 3.3 to 4.9 million, highlighting the growing disease burden of IBD globally [5]. Chronic inflammation is associated with the accumulation of tissue damage that can lead to disease complications such as strictures, fistulae, and surgical resections [1]. CD has a significant negative impact on the health-related and social quality of life of diagnosed individuals, and this includes recurring disease activity, extra-intestinal manifestations, and decreased social functioning and productivity, and has been linked to colonic cancer [6,7,8]. Up to half of individuals with CD will experience a disease complication requiring surgery within 10 years of diagnosis [1]. The Montreal Classification categorizes CD into phenotypes based on age at diagnosis, disease location, and aggressiveness of the disease [3,9,10]. Evidence suggests that some of the CD phenotypes that initially present as mild disease tend to evolve into complicated moderate to severe phenotypes which require more aggressive pharmacotherapy and surgical interventions, over time [11]. The health and social impact of CD is worse for individuals with severe phenotypes [12]. These phenotypes contribute a large proportion of difficult-to-treat CD, a subtype of CD that has limited treatment options [13].

Since 1975, biologic therapies have evolved through the decades and have become part of treatment protocols for several diseases such as IBD, rheumatoid arthritis, and some cancers [14,15]. Available treatment modalities for CD target the resolution of clinical symptoms, such as diarrhea and extraintestinal manifestations, and endoscopic remission, which includes lack of inflammation, ulceration, and bleeding in the affected areas [9,10]. According to recent guidelines, people with moderate-to-severe CD are started on either Tumor Necrosis Factor alpha (TNF-α) inhibitor monotherapy, such as infliximab and adalimumab, or combined therapy consisting of TNF-α inhibitors and immunomodulators (thiopurines, 5-aminosalicylic acid (5-ASA), methotrexate) for the induction and maintenance of remission, with the use of corticosteroids for short-term management of flare-ups [16]. However, around one-third of individuals on TNF-α inhibitors do not achieve induction of remission, and of those who achieve initial remission, around one-half fail to maintain remission subsequently [17]. For these individuals with refractory CD, treatment options are limited as surgical approaches generally do not result in long-term remission, while the repeated use of corticosteroids comes with significant risks [17].

The lack of treatment options has led to the development of interleukin 12 and 23 (IL-12/IL-23) inhibitors, another group of monoclonal antibodies, the most common being Ustekinamab [18]. IL-12/IL-23 are pro-inflammatory cytokines, sharing a similar p40 subunit, which are produced by immune cells in response to antigens [18]. The first generation of IL-12/IL-23 monoclonal antibodies, primarily the drug Ustekinamb, blocks the shared p40 subunits of the two cytokines, but increased understanding of the pathophysiology of CD has led to the development of selective anti-IL-23 monoclonal antibodies, which inhibit the p19 subunit of IL-23 while preserving IL-12 functionality [17,18]. Examples of the selective anti-IL-23 monoclonal antibodies include Mirikizumab, Risankizumab, and Guselkumab [11].

IL-12/IL-23 inhibitors are active proteins and therefore have been traditionally administered using the intravenous route [19,20]. For the treatment of CD, IL-12/IL-23 inhibitors are administered through a series of intravenous infusions [21]. Advantages of the intravenous administration of IL-12/IL-23 inhibitors include a quicker drug effect, better control of dosages, continuous drug administration, and larger dose infusion and allows drug administration in people who are unable to take drugs via oral and other routes [22]. Recent advances have resulted in formulations of IL-12/IL-23 inhibitors that can be delivered via subcutaneous injections [20]. This route enhances adherence and lowers treatment costs, as it allows self-administration, improves individual mobility, reduces hospital stay, reduces overall costs, and enables the administration of biologics in people with poor venous access [22]. Additionally, subcutaneous injections typically take minutes compared to the hours needed for intravenous infusions, which makes it a compelling and efficient alternative, favored by both individuals and healthcare providers [23,24].

Results from randomized controlled trials that have compared the efficacy of subcutaneous and intravenous administration of IL-12/IL-23 inhibitors have been inconsistent [25,26,27,28,29,30,31]. Generally, subcutaneous IL-12/IL-23 inhibitors were found to be inferior in efficacy when compared to intravenous infusion for the induction of remission [28,29,30]. One study, however, demonstrated the superiority of the subcutaneous route for the induction of CD remission [25]. For the maintenance of remission, there are only a few studies, and their findings suggest that subcutaneous administration of IL-12/IL-23 inhibitors may match or supersede the efficacy of intravenous infusions [30,31]. Given the disparity in the results of existing clinical trials, there is a need to synthesize existing evidence on the comparative efficacy of subcutaneous and intravenous routes of IL-12/IL-23 inhibitors, to enable clinicians to make informed choices with their patients. In this meta-analysis, we synthesized the evidence from existing randomized controlled trials on the efficacy and safety of the subcutaneous route of administration of IL-12/IL-23 inhibitors compared to the intravenous infusion, for CD remission.

## 2. Methods

### 2.1. Study Design

This is a systematic review and meta-analysis which was conducted according to the Preferred Reporting Items for Systematic Reviews and Meta-Analyses (PRISMA) guidelines [32]. The protocol for this study is registered in the International Prospective Register of Systematic Reviews (PROSPERO) (CRD42024484820). 

### 2.2. Data Sources

A search was conducted of Cochrane Central Register of Controlled Trials (CENTRAL), Scopus, PubMed, Cumulated Index to Nursing and Allied Health Literature (CINAHL), the databases of preprints (medRXIV), Google Scholar, ClinicalTrials.gov, and references. This search covered databases up to the 20th of September 2024, with no language restrictions. The search strategy is shown in the Appendix A.

### 2.3. Search Terms

#### 2.3.1. Search Terms for Interleukin 12/23 Inhibitors

“Monoclonal antibody” OR “Monoclonal antibodies” OR “Biologics” OR “Ustekinumab” OR “Stelara” OR “Wezlana” OR “CNTO-1275” OR “CNTO1275” OR “anti-IL-12” OR “anti-IL-23” OR “Anti-IL-12/23 Therapy” OR “IL-12/23 Blocker” OR “IL-12/23 Inhibitor” OR “Guselkumab” OR “Tremfya” OR “Risankizumab” OR “Skyrizi” OR “Brazikumab” OR “Mirikizumab” OR “Omvoh”.

#### 2.3.2. Search Terms for Crohn’s Disease

“Crohn’s Disease” OR “Crohns Disease” OR “Crohn Disease” OR “CD” OR “Crohn’s Enteritis” OR “Regional Enteritis” OR “Ileocolitis” OR “Ileitis, Terminal” OR “Terminal Ileitis” OR “Ileitis, Regional” OR “Regional Ileitides” OR “Regional Ileitis” OR “Enteritis, Granulomatous” OR “Granulomatous Enteritis” OR “Refractory Crohn’s Disease” OR “Severe Crohn’s Disease” OR “Mild-to-Moderate Crohn’s Disease”.

#### 2.3.3. Search Terms for Route of Administration

“Subcutaneous” OR “SC” OR “SQ” OR “SubQ” OR “Sub-Q” OR “Subcut” OR “SubC” OR “Subcu” OR “Hypodermic” OR “Hypodermal” OR “Intracutaneous” OR “Intravenous” OR “Drip” OR “Endovenous” OR “IV” OR “Venous” OR “Infusion”

### 2.4. Procedure for the Selection of Studies

The study records obtained from the literature search were imported into EndNote 20 software for de-duplication. Subsequently, the records were uploaded to the Rayyan systematic review management website (https://www.rayyan.ai/ (accessed on 17 August 2024)) for screening based on the title and abstract. Preliminarily included study records were then retrieved in full text and manually assessed for eligibility by two independent investigators. In cases of disagreement, a third investigator was consulted to make the final decision.

### 2.5. Eligibility

Studies were eligible for inclusion if they were RCTs evaluating the efficacy and safety of subcutaneous compared to intravenous IL-12/IL-23 inhibitors in participants with CD. We excluded observational studies and trials where the comparator was placebo. This approach was adopted to ensure that only studies with a direct, head-to-head comparison between subcutaneous and intravenous administration of IL-12/IL-23 inhibitors in CD were included.

### 2.6. Outcomes

The primary efficacy outcomes were the induction and maintenance of remission, which were defined by the Crohn’s Disease Activity Index (CDAI) as asymptomatic remission <150 or a Harvey Bradshaw Index (HBI) of ≤4 points [33]. The secondary efficacy outcomes were endoscopic remission, defined as a Simple Endoscopic Score for Crohn’s Disease (SES-CD) of 0–2, and endoscopic response, defined as a ≥50% reduction from baseline in SES-CD [34]. The primary safety outcome was any adverse events. The secondary safety outcomes were all-cause mortality, treatment discontinuation, and serious adverse events.

### 2.7. Data Extraction

Data were extracted from the included studies on Microsoft Excel. Eight authors (NA, RI, HA, K-HA, M-AA, N-AA, W-NA, YA) independently extracted data on the characteristics such as study design, date, location, number of participants, and selected demographic characteristics (e.g., age, gender, comorbidities, type, and severity of CD). Data regarding the treatment regimen included the drugs used, the dosages, and the duration of treatment. Data related to the study outcome included the number of individuals in both the subcutaneous and intravenous groups who achieved induced remission, the number of participants in these groups who maintained remission, the number of individuals who developed at least one adverse event, and the number of individuals that developed at least one serious adverse event, death, or treatment discontinuation. 

### 2.8. Assessing the Quality of Included Studies

The quality of the included studies was evaluated using the Methodological Standard for Epidemiological Research (MASTER) scale, comprising seven domains subdivided into 36 safeguards [35]. These domains are to assess the internal validity of included studies including selection bias, information bias, confounding, and bias in the study design and in the analysis and interpretation of results.

### 2.9. Data Synthesis

Characteristics of included studies are displayed in tables and narratively described. For outcomes (maintenance of remission and endoscopic remission) where there were insufficient studies for meta-analysis, results are described narratively. A meta-analysis was carried out for induction, endoscopic response, mortality, and safety outcomes using a bias-adjusted meta-analysis model—the quality-effects model [36]—which is a model under the common parameters assumption. The study level unadjusted odds ratios (ORs) and their 95% CIs were recalculated for each of the binary outcomes and then synthesized to compute overall effects as this is considered the optimal binary measure in meta-analysis [37]. The quality-effects model modifies variance weights using the quality ranking of the studies, thereby decreasing bias in the synthesized results [38]. Sensitivity analysis was conducted using leave-one-out analysis. We used the I^2^ statistic, Cochran’s Q *p*-value, and tau^2^ to quantify heterogeneity, and I^2^ values of 25%, 50%, and 75% were assigned into low, moderate, and high inconsistency categories, respectively [39,40]. We investigated publication bias using Doi plots and the LFK index [41]. GRADE was used to report the evidence for each outcome. We used Stata version 18 (College Station, TX, USA) with the *metan* module in Stata.

### 2.10. Ethics

This systematic review utilized published data; therefore, there was no need for ethical approval.

## 3. Results

### 3.1. Search Results

A total of 3287 studies were identified through the search. After eliminating duplicates and a manual screening process based on the title and abstract, 37 studies underwent full-text screening. Out of these, 30 studies were excluded for the reasons indicated in Figure 1. After the full-text screening, most of the studies were excluded because they did not compare the administration routes (*n* = 12). Ultimately, seven trials were included [25,26,27,28,29,30,31], four of them were included in the induction analysis [25,28,29,30], two studies were included in the maintenance analysis [30,31], and two trials did not provide data on either induction or maintenance [26,27]. Further, seven RCTs [25,26,27,28,29,30,31] were included in the analysis of adverse events and serious adverse events [25,26,27,28,29,30,31], five trials in the treatment discontinuation analysis [25,28,29,30,31], and six trials in the mortality analysis [25,26,27,28,29,31].

### 3.2. Characteristics of Included Studies

The seven selected studies [25,26,27,28,29,30,31] involved 2179 participants from more than 40 countries with most studies from the United States, Canada, and Germany. All seven trials included participants with moderate to severe CD [25,26,27,28,29,30,31]. Three of the studies [28,29,30] investigated subcutaneous Ustekinumab, two investigated Risankizumab [26,27], one investigated subcutaneous Mirikizumab [31], and one investigated Brazikumab [25]. However, the two studies [26,27] that compared subcutaneous to intravenous Risankizumab only reported data on adverse events. The characteristics of the included studies are shown in Table 1. 

### 3.3. Assessment of the Quality of Included Studies

Most of the included trials obtained MASTER scale scores ranging from 32 to 35 out of 36, with an average of 33.5, signifying an overall high quality of evidence (Appendix A). The temporal precedence domains were met by all the studies except one [28] because of the possible presence of carry-over effects. All the trials were multicenter trials and therefore the safeguard of equal care delivery was not met.

### 3.4. Induction of Remission

Four RCTs [25,28,29,30] compared subcutaneous to intravenous IL-12/IL-23 inhibitors for the induction of remission in CD. Three of these trials [28,29,30] investigated subcutaneous Ustekinumab and the remaining one studied Brazikumab [25]. In the overall synthesis, the odds of inducing remission were reduced by almost 23% in the subcutaneous group compared to the intravenous group (OR 0.77, 95%CI 0.53–1.12), with the confidence interval suggesting that the treatments could be similar, with no-to-low heterogeneity (I^2^ = 0%) (Figure 2) and major asymmetry (Appendix A), suggesting concerns with publication bias. The Galbraith plot (Appendix A) showed an effect near the null, represented by an almost horizontal line. The results were consistent after sensitivity analysis using leave-one-out (Appendix A), and after restricting the analysis to Ustekinumab only. The summary GRADE rating for this outcome was of moderate-certainty evidence (Appendix A).

### 3.5. Maintenance of Remission

Two RCTs [30,31] compared subcutaneous to intravenous IL-12/IL-23 inhibitors for maintaining CD remission. The first trial [30] showed nearly 40% higher odds in achieving remission with subcutaneous compared to intravenous Ustekinumab in weeks 12 and 16, while week 14 showed nearly the same odds of maintaining remission in both routes. The second trial [31] showed higher odds of maintaining remission when subcutaneous Mirikizumab was used compared to the intravenous route (Appendix A).

### 3.6. Endoscopic Response and Remission

Three RCTs [28,29,31] reported endoscopic response, and the overall odds ratio and its 95%CI suggested no difference in the odds of achieving an endoscopic response with the subcutaneous administration compared to the intravenous administration, with no-to-low heterogeneity (OR 0.73, 95%CI 0.46–1.15, I^2^ = 0%) and no asymmetry in the Doi plots, suggesting no publication bias (Appendix A). The summary GRADE rating for endoscopic response was of high-certainty evidence (Appendix A). Two RCTs [28,31] reported endoscopic remission (Appendix A). The first trial [28] showed a 76% decrease in the odds of achieving endoscopic remission using subcutaneous compared to intravenous Ustekinumab (OR 0.24, 95%CI 0.06–0.90). The second trial [31] showed two-fold higher odds of achieving endoscopic remission using subcutaneous compared to intravenous Mirikizumab (OR 2.00, 95%CI 0.74–5.36). 

### 3.7. Any Adverse Events and Serious Adverse Events

A total of seven RCTs reported adverse events [25,26,27,28,29,30,31], and the overall effect estimate showed similar odds of adverse events with subcutaneous administration compared to intravenous, with low heterogeneity (OR 0.91, 95%CI 0.63–1.32, I^2^ = 39%) and no asymmetry in the Doi plot, suggesting no publication bias (Figure 3A, Appendix A). Data from the same seven RCTs [25,26,27,28,29,30,31] reported the serious adverse events, with the overall effect showing nearly the same odds of serious adverse events, with no-to-low heterogeneity (OR 0.97, 95%CI 0.61–1.53, I^2^ = 0%) and minor asymmetry in the Doi plot suggesting little-to-no publication bias (Figure 3B, Appendix A). The summary GRADE rating for both outcomes was of high-certainty evidence (Appendix A).

#### 3.7.1. Treatment Discontinuation

Overall, five RCTs [25,28,29,30,31] reported data on treatment discontinuation, and the overall effect estimate showed almost similar odds of treatment discontinuation with subcutaneous compared to intravenous administration, with no-to-low heterogeneity (OR 1.06, 95%CI 0.67–1.68, I^2^ = 0%) (Figure 4, Appendix A). The summary GRADE rating for this outcome was of moderate-certainty evidence due to possible publication bias (Appendix A). The most common reasons for discontinuation across both groups were due to lack of efficacy and voluntary withdrawal (Appendix A).

#### 3.7.2. Hospitalization

One RCT [29] reported the number of hospitalizations, and the results suggested similar odds of hospitalization in the subcutaneous arm compared to the intravenous arm (OR 0.97, 95%CI 0.04–24.08). 

#### 3.7.3. Mortality

Out of the included RCTs, six reported the mortality outcome [25,26,27,28,29,31]. The overall synthesis showed almost similar odds of mortality in the subcutaneous compared to the intravenous route, with no-to-low heterogeneity (OR 1.17, 95%CI 0.27–5.20, I^2^ = 0%) (Figure 5) and no evidence of publication bias (Appendix A). The summary GRADE rating for this outcome was of high-certainty evidence (Appendix A).

## 4. Discussion

In this meta-analysis of seven RCTs, we found that subcutaneous administration of IL-12/IL-23 inhibitors, compared to intravenous administration, was associated with a small non-significant decrease in efficacy when used for induction of remission in individuals with CD. As for the safety of subcutaneous IL-12/IL-23 inhibitors, similar odds of adverse events were observed compared to intravenous IL-12/IL-23 inhibitors. However, for the maintenance of remission, data from two trials suggested that the subcutaneous route was similar to the intravenous in efficacy. Additionally, the subcutaneous route showed almost similar odds of mortality compared to the intravenous route.

We found that subcutaneous administration showed no difference in the odds of induction of remission in individuals with CD, with low-certainty GRADE evidence. The one existing meta-analysis comparing the subcutaneous and intravenous administration of IL-12/IL-23 inhibitors for the induction of CD remission used observational studies, which is associated with key methodological limitations [42], including an indirect comparison of the two administration routes across studies that were performed separately for each of the routes. This lacks the rigor of a head-to-head trial, as the study populations were different. While the analysis concluded that subcutaneous Ustekinumab was superior in inducing a clinical response, the conclusions are not based on high-quality data and analysis. Notably, current guidelines do not provide explicit recommendations regarding the subcutaneous route for the induction phase, due to the lack of robust evidence regarding the efficacy of subcutaneous interleukin inhibitors compared to the intravenous route, which this meta-analysis now addresses.

Our findings suggest that subcutaneous administration of IL-12/IL-23 inhibitors had similar odds of maintaining remission compared to the intravenous route. However, our findings are limited as only two RCTs have so far been conducted and have provided data for maintaining remission. One of the two RCTs [30] showed that subcutaneous Ustekinumab increased the odds of maintaining remission by 40% and 44% in weeks 12 and 16, respectively. The second included RCT [31] and showed a two-fold increase in the odds of maintaining remission in subcutaneous compared to intravenous Mirikizumab at 52 weeks. In both trials, the 95%CIs include the null, suggesting that the two routes could be similar. There are no existing meta-analyses that compare the two routes of administration of IL-12/IL-23 inhibitors in maintaining remission. Nonetheless, some meta-analyses have shown that subcutaneous administration of anti-TNF-α monoclonal antibodies, which are another class of biologics commonly prescribed to individuals with CD, may result in similar efficacy to intravenous infusions. Subcutaneous adalimumab showed similar efficacy in maintaining remission in individuals with CD compared to intravenous Infliximab in one published meta-analysis [43]. The same has been observed for subcutaneous infliximab when compared to intravenous infliximab or subcutaneous/intravenous Vedolizumab in another meta-analysis of 13 RCTs, although this study had limitations of indirect comparisons [44]. Despite the different mechanism of action between these drugs and IL 12/23 inhibitors, the findings support the results of subcutaneous IL 12/23 inhibitors in maintaining remission in the tow studies in this current meta-analysis. Although there are multiple available IL-12/IL-23 inhibitors, the current European Crohn’s and Colitis Organization (ECCO) guidelines recommend the use of subcutaneous Ustekinumab to maintain clinical remission in individuals with CD who achieved remission with Ustekinumab [45]. This might be due to the lack of data on subcutaneous formulations of other monoclonal antibodies. For example, there are two RCTs, the ADVANCE and MOTIVATE trials, which initially set out to compare subcutaneous to intravenous Risankizumab, in addition to intravenous vs. placebo [46]. However, the two trials have not reported the efficacy findings of the comparison for subcutaneous and intravenous administration of this selective IL-23 blocker and data on safety comparisons used in our meta-analysis were only available on ClinicalTrials.gov [46].

Our current meta-analysis found that adverse events and treatment discontinuation were almost similar between subcutaneous and intravenous IL-12/IL-23 inhibitors. Again, there are no previous meta-analyses that compare either adverse events or drug discontinuation between the two routes of administration, both of which are key factors in influencing outcomes, especially in individuals who are refractory to other therapies. Although there was a slight increase in the odds of mortality in the subcutaneous group, the confidence interval suggested that there was no clinically important difference in mortality between the two routes of administration. Our findings suggest that it is equally safe to administer IL-12/IL-23 inhibitors via the subcutaneous route as compared to intravenous administration, although more trials are still required. A similar safety profile is important for individuals on biologic treatment, as many report preferring intravenous over subcutaneous administration because they feel safer in a hospital setting [47]. Thus, with a similar safety profile, providers can reassure individuals that the subcutaneous formulation is as safe as the intravenous one, and that they can return to the hospital if they feel any side-effects.

The strength of this meta-analysis includes the comprehensive search and rigorous analysis that was carried out, and the use of GRADE evidence of certainty makes the results easier for translation into practice guidelines and clinical practice. To the best of our knowledge, this study provides the only direct comparison between subcutaneous and intravenous IL-12/IL-23 inhibitors administration using data from RCTs, both of which are available for induction and maintenance of remission. A limitation of this study is that there were insufficient data on the maintenance of remission in CD, limiting our ability to draw reliable conclusions. Limitations of the included studies include the lack of consistent and objective measures of remission, the variable induction timing between the studies, the lack of classification of disease severity, and the lack of data on the use of rescue steroids during the trial period in most studies. Additionally, two trials, the ADVANCE [46] and the MOTIVATE [46], have published data on just the intravenous versus placebo comparisons and therefore could not be included in the efficacy outcomes. More trials that compared the subcutaneous and intravenous routes for IL-12/IL-23 inhibitors are needed to improve evidence certainty.

## 5. Conclusions

In individuals with moderate to severe CD, subcutaneous administration of IL-12/IL-23 inhibitors has similar efficacy to intravenous administration for the induction and maintenance of CD remission and safety. However, for robust conclusions about the use of this route for IL12/23 inhibitors for the maintenance of CD remission, more studies are needed. 

## Figures and Tables

**Figure 1 biomedicines-13-00702-f001:**
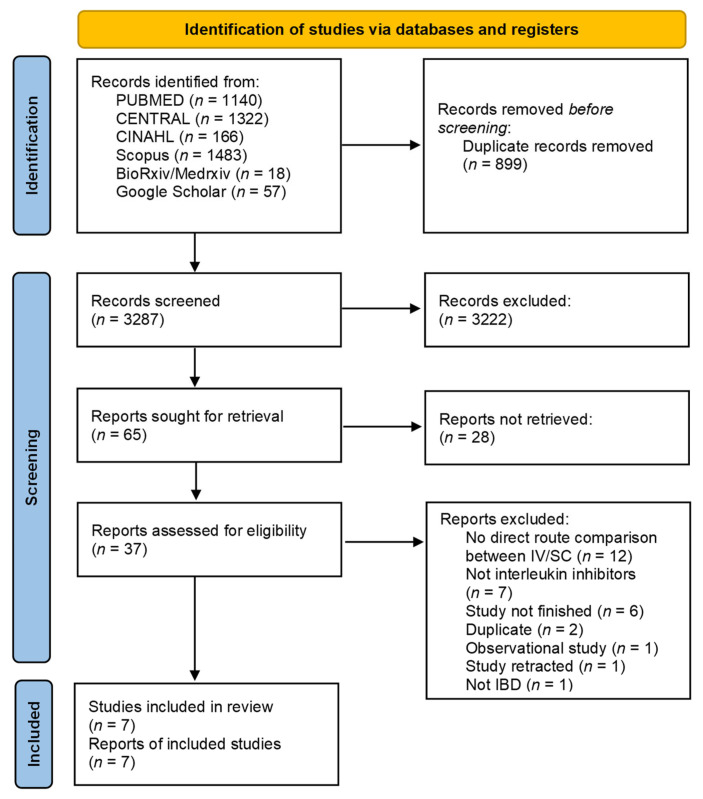
PRISMA flow diagram.

**Figure 2 biomedicines-13-00702-f002:**
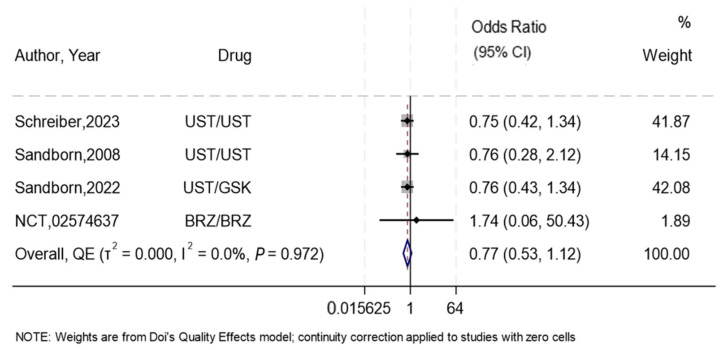
Forest plot showing efficacy of subcutaneous compared to intravenous interleukin 12/23 inhibitors for induction of remission. NB: This forest plot shows the overall analysis for the odds of inducing remission using the subcutaneous compared to the intravenous route in four RCTs [25,28,29,30]. The figure includes a line of no effect (vertical black line), meta-analytical estimates with confidence intervals (blue diamonds), study weights (gray boxes), study effect sizes (black dots), confidence intervals for study effect sizes (horizontal black lines), and a line representing the overall odds ratio (red dotted line). Abbreviations: Ustekinumab (UST), Guselkumab (GSK), and Brazikumab (BRZ).

**Figure 3 biomedicines-13-00702-f003:**
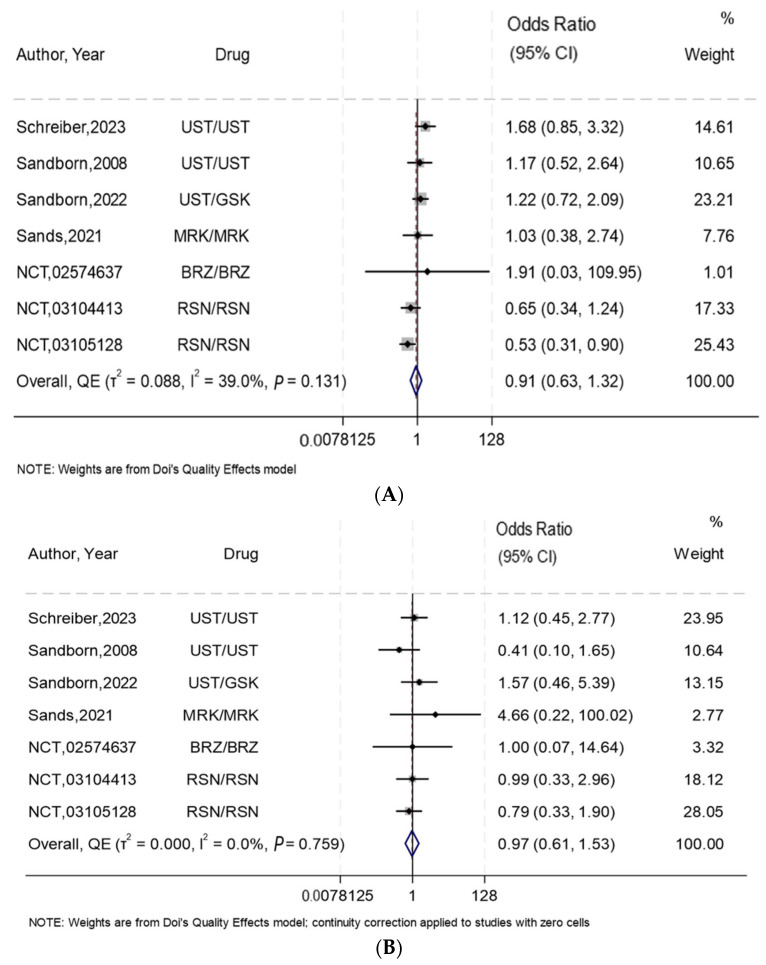
Forest plots showing adverse events and serious adverse events of subcutaneous compared to intravenous interleukin 12/23 inhibitors. NB: This forest plot shows the analysis of the safety outcomes of (**A**) the development of adverse events and (**B**) serious adverse events in individuals with CD using subcutaneous compared to the intravenous interleukin 12/23 inhibitors in seven RCTs [25,26,27,28,29,30,31]. The figure includes a line of no effect (vertical black line), meta-analytical estimates with confidence intervals (blue diamonds), study weights (gray boxes), study effect sizes (black dots), confidence intervals for study effect sizes (horizontal black lines), and a line representing the overall odds ratio (red dotted line). Abbreviations: Ustekinumab (UST), Guselkumab (GSK), Brazikumab (BRZ), Mirikizumab (MRK), and Risankizumab (RSN).

**Figure 4 biomedicines-13-00702-f004:**
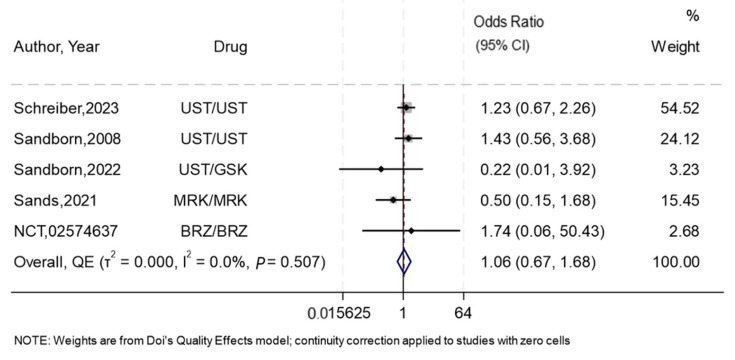
Forest plot for the effect of subcutaneous compared to intravenous interleukin 12/23 inhibitors on treatment discontinuation. NB: This forest plot shows the overall analysis for the odds of treatment discontinuation with the subcutaneous compared to the intravenous route using IL12/23 inhibitors in five RCTs [25,28,29,30,31]. The figure includes a line of no effect (vertical black line), meta-analytical estimates with confidence intervals (blue diamonds), study weights (gray boxes), study effect sizes (black dots), confidence intervals for study effect sizes (horizontal black lines), and a line representing the overall odds ratio (red dotted line). Abbreviations: Ustekinumab (UST), Guselkumab (GSK), Brazikumab (BRZ), and Mirikizumab (MRK).

**Figure 5 biomedicines-13-00702-f005:**
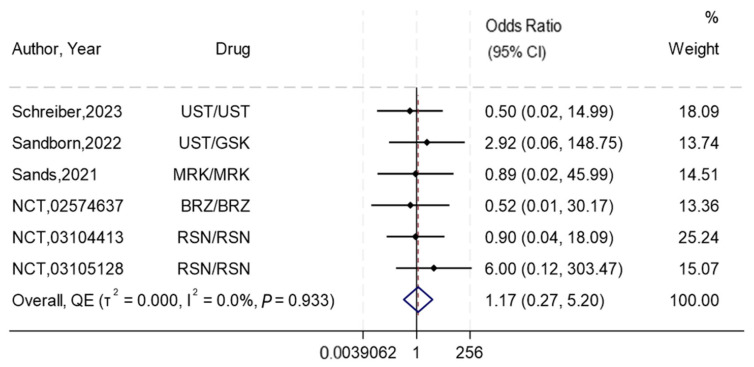
Forest plot for the effect of subcutaneous compared to intravenous interleukin 12/23 inhibitors on mortality. NB: This forest plot shows the overall analysis for the odds of mortality using the subcutaneous compared to the intravenous route in six RCTs [25,26,27,28,29,31]. The figure includes a line of no effect (vertical black line), meta-analytical estimates with confidence intervals (blue diamonds), study weights (gray boxes), study effect sizes (black dots), confidence intervals for study effect sizes (horizontal black lines), and a line representing the overall odds ratio (red dotted line). Abbreviations: Ustekinumab (UST), Guselkumab (GSK), Brazikumab (BRZ), and Mirikizumab (MRK).

**Table 1 biomedicines-13-00702-t001:** Characteristics of included studies.

Author, Year	Type of IBD	Specific Biologic	SC/IV Sample Size	Phase of Remission	Follow-Up Period (Months)	Mechanism of Action
Schreiber, 2023 [28] (NCT,03782376)	Crohn’s Disease	SC: Ustekinumab: 90 mg IV: Ustekinumab: 6 mg/kg	SC:107 IV:108	Induction	8	IL-12/IL-23 Inhibitor
Sandborn, 2008 [30] (NCT,00265122)	Crohn’s Disease	SC: Ustekinumab: 90 mg IV: Ustekinumab: 4.5 mg/kg	SC:65 IV:66	Induction and Maintenance	5.22	IL-12/IL-23 Inhibitor
Sandborn, 2022 [29] (NCT,03466411)	Crohn’s Disease	SC: Ustekinumab: 90 mg IV: Guselkumab: 200 mg, 600 mg, 1200 mg	SC:63 IV:185	Induction	2.78	IL-12/IL-23 Inhibitor
Sands, 2021 [31] (NCT,02891226)	Crohn’s Disease	SC: Mirikizumab: 300 mg IV: Mirikizumab: 1000 mg	SC:46 IV:41	Maintenance	12	IL-23 Inhibitor
NCT,02574637 [25]	Crohn’s Disease	SC: Brazikumab: 35, 70, 105, 210 mg IV: Brazikumab: 700 mg	SC:10 IV:5	Induction	6.4	IL-23 Inhibitor
NCT,03104413 [26]	Crohn’s Disease	SC: Risankizumab: 180, 360 mg IV: Risankizumab: 1200 mg	SC: 83 IV: 453	Adverse events	10.1	IL-23 Inhibitor
NCT,03105128 [27]	Crohn’s Disease	SC: Risankizumab: 180, 360 mg IV: Risankizumab: 1200 mg	SC:135 IV:812	Adverse events	10.1	IL-23 Inhibitor

## Data Availability

This is a meta-analysis. Data are included in primary studies according to policies of parent journals.

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
