# Peer review of "Comparative Efficacy of Subcutaneous Versus Intravenous Interleukin 12/23 Inhibitors for the Remission of Moderate to Severe Crohn’s Disease: A Systematic Review and Meta-Analysis"

_biomedicines, 2025, doi:10.3390/biomedicines13030702_

Round 1

Reviewer 1 Report

Comments and Suggestions for Authors

The authors have reviewed as well as presented their meta analysis

The manuscript needs to be revised

1. Sometimes its hard to understand if it is a review paper or a research paper. 

2. looking at the meta analysis it looks like a research paper

3. From a review paper perspective, there is not much of review that has been done. More literature survey is needed. Kindly enhance

4. Clarify the article type. 

5. Figure legends need to be more self explanatory

6. Abstract has sub sections, is that needed? refere Biomedicines style

7. Discussion is a mess, kindly rewrite

8. Conclusions ?

9. the last para of introduction highlightes the novelty of the study, nothing of that sort is there, kindly add the objective and highlights of the study in the last para. 

Language issues need polishing. 

Comments on the Quality of English Language

minor

Author Response

Response to reviewers

Dear Editor, we are grateful to the journal and the reviewers for all the suggestions and comments. Below, we have responded to each suggestion, and we have also made changes in the revised manuscript, to address these comments.

Reviewer #1:

  1. Sometimes it’s hard to understand if it is a review paper or a research paper. 

We thank the reviewer for this comment. This is a systematic review and meta-analysis. It is likely that the confusion in the article type may come from the journal labels meta-analyses under “Review” during peer-review.

  1. looking at the meta-analysis it looks like a research paper

We thank the reviewer for this comment.

  1. From a review paper perspective, there is not much of review that has been done. More literature survey is needed. Kindly enhance.

We thank the reviewer for this comment. However, this is not a literature review, as stated in prior responses.

  1. Clarify the article type. 

We thank the reviewer for this comment. This is a systematic review and meta-analysis.

  1. Figure legends need to be more self-explanatory

We thank the reviewer for this comment. More information has been added to the legends.

  1. Abstract has sub sections, is that needed? refere Biomedicines style

We thank the reviewer for this comment. The biomedicine style should contain the following headings: Background/Objectives, Methods, Results, and Conclusions.

  1. Discussion is a mess, kindly rewrite

We thank the reviewer for this comment. The discussion has been rewritten.

  1. Conclusions?

We thank the reviewer for this comment. We have also added the conclusion in both the abstract and the main text.

"In individuals with moderate to severe CD, subcutaneous administration of IL 12/23 inhibitors is similar to intravenous administration for the induction of remission and safety, but to draw robust conclusions about the maintenance phase, more studies are needed. These findings show that the similarity between subcutaneous and intravenous routes for IL 12/23 inhibitors could influence current guidelines to consider subcutaneous formulations as an alternative for inducing remission in CD patients."

  1. the last para of introduction highlightes the novelty of the study, nothing of that sort is there, kindly add the objective and highlights of the study in the last para

We thank the reviewer for this comment. We have added it.

"Given the disparity in the results of existing clinical trials, there is a need to synthesize the existing evidence on the comparative efficacy of subcutaneous and intravenous routes of IL-12/IL-23 inhibitors, to enable clinicians to make informed choices with their patients. The novelty of this study lies in its comprehensive analysis of head-to-head comparisons between the two administration routes, offering clarity on their relative clinical effectiveness and safety. The objective was to address the existing knowledge gap and provide robust evidence to guide treatment strategies. In this meta-analysis, we synthesized the evidence from existing randomized controlled trials on the efficacy and safety of the subcutaneous route of administration of IL-12/IL-23 inhibitors to the intravenous infusion."

Reviewer 2 Report

Comments and Suggestions for Authors

The work by Alwisi et al. presents a meta-analysis of papers that investigated the efficacy of subcutaneous versus intravenous interleukin 12/13 inhibitors. The paper is written clearly, concisely and transparently. The introduction contains all the necessary relevant information as well as the goal of this meta-analysis. However, minor revisions are required before the publication. It is necessary to go through the manuscript and standardize the font, letter size, spacing as well as excess spaces throughout the manuscript.

When listing authors, the number of the affiliation should be written in superscript - please correct it.

Also, affiliation should be numbered to know which affiliation is number 1 and which is number 2.

Abstract is too long. It should contain a maximum of 250 words, and you have over 300 in your paper.

Introduction

For these individuals with refractory CD, treatment options are limited as surgical approaches generally do not result in long term remission, while the repeated use of corticosteroids comes with significant risks (17). - redo the bracket for this reference.

The first-genera-tion of IL-12/IL-23 monoclonal antibodies, primarily the drug Ustekinamb, block the shared p40 subunits of the two cytokines but increased understanding of the pathophys-iology of CD has led to the development of selective anti-IL-23 monoclonal antibodies, which inhibit the p19 subunit of IL-23 while preserving IL-12 functionality (17, 18). - redo the bracket for this references.

The discussion should be refined a little more.

6 Declarations is unnecessary - feel free to drop it.

CRediT Authors’ contributions - revise and write it better. See journal’s instructions.

References are not written according to the journal's instructions. Please correct it.

Author Response

Response to reviewers

Dear Editor, we are grateful to the journal and the reviewers for all the suggestions and comments. Below, we have responded to each suggestion, and we have also made changes in the revised manuscript, to address these comments.

Reviewer #2:

  1. When listing authors, the number of the affiliation should be written in superscript - please correct it.

We thank the reviewer for this comment. The affiliations have been adjusted.

  1. Also, affiliation should be numbered to know which affiliation is number 1 and which is number 2.

We thank the reviewer for this comment. The affiliations have been adjusted.

  1. Abstract is too long. It should contain a maximum of 250 words, and you have over 300 in your paper.

We thank the reviewer for this comment. The abstract has been shortened to make it more concise and clearer.

  1. For these individuals with refractory CD, treatment options are limited as surgical approaches generally do not result in long term remission, while the repeated use of corticosteroids comes with significant risks (17). - redo the bracket for this reference.

We thank the reviewer for this comment. The brackets have been corrected.

  1. The first-genera-tion of IL-12/IL-23 monoclonal antibodies, primarily the drug Ustekinamb, block the shared p40 subunits of the two cytokines but increased understanding of the pathophys-iology of CD has led to the development of selective anti-IL-23 monoclonal antibodies, which inhibit the p19 subunit of IL-23 while preserving IL-12 functionality (17, 18). - redo the bracket for this references.

We thank the reviewer for this comment. The brackets have been corrected.

  1. The discussion should be refined a little more.

We thank the reviewer for this comment. The discussion has been rewritten.

  1. 6 Declarations is unnecessary - feel free to drop it.

We thank the reviewer for this comment. The declaration section has been adjusted.

  1. CRediT Authors’ contributions - revise and write it better. See journal’s instructions.

We thank the reviewer for this comment. The contributions have been revised and rewritten.

  1. References are not written according to the journal's instructions. Please correct it

We thank the reviewer for this comment. The references have been corrected.

Round 2

Reviewer 1 Report

Comments and Suggestions for Authors

Accept